



# Brief communication: Impact of mountain glaciers on regional hydroclimate

Husile Bai[1*], Summer Rupper[2], and Courtenay Strong[3]

[1]Department of Earth and Environmental Sciences, Vanderbilt University, Nashville, Tennessee
[2]School of Environment, Society, and Sustainability, University of Utah, Salt Lake City, Utah
[3]Department of Atmospheric Sciences, University of Utah, Salt Lake City, Utah

**Correspondence:** Husile Bai (husile.bai@vanderbilt.edu)

**Abstract.** The crucial role of glaciers as a water supply underscores the need to reliably simulate alpine climate change, including glacier-atmosphere interactions. The presence of a glacier can change precipitation by generating mountain-valley scale flows, but we show here that their impacts on the atmosphere are more profound and much larger in scale. In a validated regional climate model, modest changes to the size of glacier termini in the Karakoram altered the large-scale summer monsoonal
5 circulation, producing precipitation anomalies of sufficient magnitude and scale to overwhelm valley-scale orographic effects. Notably, the robust synoptic-scale moisture flow response exerted a substantial influence on precipitation and overwhelmed the localized response of the orographic flows, highlighting the significant impact of glacier ice on the monsoonal circulation and, hence, precipitation. These changes in turn impact glacier mass balance over the Karakoram range, emphasizing the importance of proper specification of glaciated area for the study of hydroclimate monitoring.

## 1 Introduction

Glaciated mountains, often known as "water towers", supply and store water critical for environmental and human water demands downstream (Immerzeel et al., 2020). These water towers are also studied as life zones which provide sustainable resources for mountain ecosystems that represent almost one-third of the world's terrestrial ecosystem diversity, especially the plant diversity (Körner et al., 2011). The melt of glaciers is observed to transpire across a significant portion of the annual cycle,
including select winter months in diverse regions, thereby contributing vital water resources crucial for sustaining ecosystems during dry seasons, in arid regions, and amid periods of drought (Pritchard, 2019; Immerzeel et al., 2020). With glaciers experiencing widespread thinning and retreat due to climate shifts on a global scale, extensive research has been dedicated to understanding the implications of these transformations on environmental and human systems both within and downstream of glaciated mountain regions (Adler et al., 2019; Huss et al., 2008; Immerzeel et al., 2013; Rounce et al., 2023; Laurent
et al., 2020). However, glaciers also play a pivotal role in the energy budget of mountain systems. Their cold surfaces and higher albedo directly impact the energy balance at the surface, resulting in large gradients in atmospheric mass, energy, and momentum within these mountain systems.

While far less work has focused on understanding the influence of glaciers on the atmosphere, there are multiple studies that identify glacier-atmosphere interactions as potentially significant. For example, studies (Goger et al., 2022; Lin et al., 2021)





show that glaciers generate mountain-valley scale flows that alter orographic precipitation patterns, and Salerno et al. (2023) show that glaciers can significantly impact local temperature trends by modifying katabatic winds. Importantly, these processes feed back into glacier surface processes. Sauter and Galos (2016) illustrated how excluding glaciers from the atmospheric forcing data results in significant biases in energy and mass fluxes for three small glaciers in northern Italy (Sauter and Galos, 2016). Additionally, recent work has demonstrated that incorporating a debris-covered glacier category in the WRF model

significantly improves near-surface temperature and humidity simulations over the Dudh Koshi River Basin, highlighting the necessity of accurately representing glacier heterogeneity in high-resolution regional climate models (Potter et al., 2021). Together, these studies highlight the importance of glaciers on the valley-scale response of the atmosphere to the presence of glaciers and how these feed back onto the glacier mass and meltwater flux. Less is known about how glaciers project onto larger-scale regional flows, and how the presence of glaciers and valley-scale changes combine to influence regional scale

atmospheric circulation and precipitation distribution.

    Real-time operational mesoscale Numerical Weather Prediction (NWP) model forecast products, such as those provided by National Centers for Environmental Prediction (NCEP) and European Centre for Medium-Range Weather Forecast (ECMWF), often provide spatial resolutions that are too coarse for application in fine-scale glacier-related research. Although NWP models have been run successfully with grid resolutions of less than 1 km in complex terrain with specific modifications to Planetary

Boundary Layer (PBL) schemes or in Large Eddy Simulation (LES) mode (Ching et al., 2014; Seaman et al., 2012; Chow and Street, 2009; Goger et al., 2022), few frameworks have been configured to properly account for the presence of glaciers in large mountain ranges, which are often underrepresented in climate models and remote sensing datasets. Collier et al. (2013) utilized high-resolution models to explore atmospheric interactions over the Karakoram glaciers, while (Collier et al., 2015) examined the influence of debris cover on glacier ablation and associated feedbacks. However, these studies primarily focus on

near-surface layers and the glacier-atmosphere interface, with less attention given to how these localized interactions impact larger-scale hydroclimate patterns or synoptic-scale circulations.

    Critical gaps thus remain in understanding how glacier-atmosphere processes influence regional-scale atmospheric circulation and precipitation distribution, especially in complex glaciated terrains. Here, we examine how atmospheric circulations respond to glaciated complex terrain at multiple scales by using Weather Research & Forecasting (WRF) model in a nested

domain configuration, featuring default glaciers and idealized, but more representative extensions of glacier termini.

## 2   Methods

In this study, we used the advanced Weather Research & Forecasting model (Skamarock et al., 2019) version of 4.5 (WRF v4.5) with three nested domains with two-way coupling. Here, the three domains were reduced in scale at a ratio of 1:3 to focus on the Karakoram region. The innermost domain (Fig. 1a) had 0.9-km resolution and was nested in two larger domains

with 2.8-km and 8.5-km resolutions.

    Salient discrepancies are common between glaciers observed by satellites (Fig. 1b) and pixels classified as ice or snow by default in regional climate models (Fig. 1c). Working with our previously validated configuration of a regional climate model





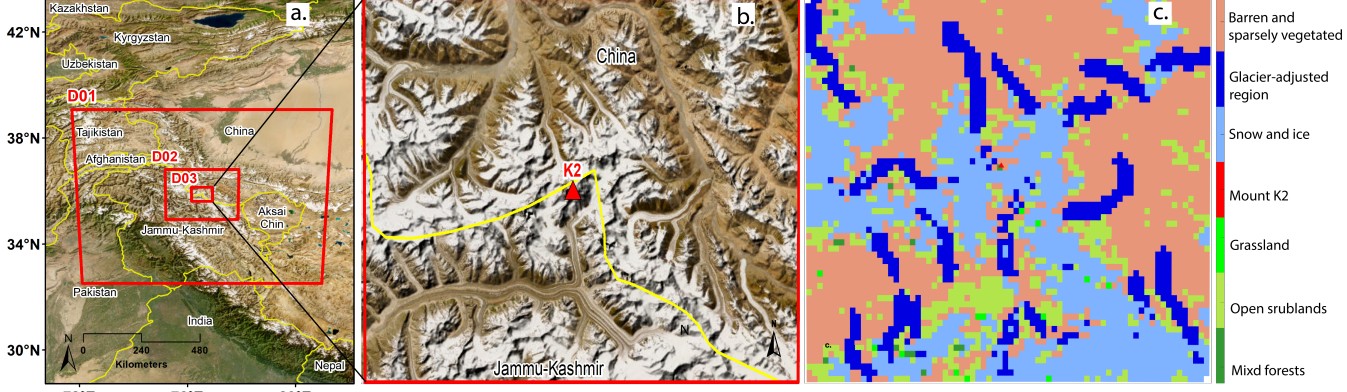

**Figure 1.** (a) WRF model domains (D01, D02, and D03) configured with horizontal spatial resolution of 8.5, 2.6, and 0.9 km. Comparison between ice and snow (b) viewed by MODIS satellite and (c) specified in a regional model's default land use (light blue). Dark blue indicates grid boxes we converted to snow and ice to better reflect missing glacier termini, and the mapped area corresponds to the innermost (0.9-km resolution) simulation domain.

(Dars et al., 2020; Wolvin et al., 2024), we apply modest changes to incorporate missing glacier termini in a portion of the Karakoram region (Fig. 1). The Karakoram was selected as a test region for four primary reasons: (1) glacier change has been extremely low compared to global averages, making it an anomaly (Shean et al., 2020); (2) glaciers are a significant water resource, serving as the headwaters to the Indus and Amu Darya (Immerzeel et al., 2020); (3) models (atmospheric, hydrological, glaciological, etc) are heavily relied upon due to extremely sparse in situ observations (Farinotti et al., 2020); and (4) precipitation is influenced by both winter westerly disturbances and summer monsoonal systems (Riley et al., 2021).

Lower-elevated valley regions were selected for addition of idealized glacier extensions by analyzing the WRF terrain with the TopoToolbox (Schwanghart and Scherler, 2014). This modification was performed in the WRF innermost domain (D03) based on the digital elevation model (DEM) in the WRF default settings, which uses the GMTED2010 30-arc-second dataset. We then analyzed the topography structure using TopoToolbox and delineated valley areas, where the elevation of the center line was the lowest (lower than the elevation in the vicinity area) and the width of valley was not exceeding five WRF D03 grid cells, yielding what we refer to as the glacier-adjusted regions (white shading in Fig. 2).

A control run was produced with default land use, and an experiment was conducted changing land use to snow/ice in selected grid boxes (Fig. 1b) to better represent, in an idealized sense, glacier termini that are visible from satellite (Fig. 1a). The model run period extended from February 1 to August 1, 2020, and July was analyzed here to illustrate effects during the summer monsoon. The configuration of the WRF model is presented in the Table 1.





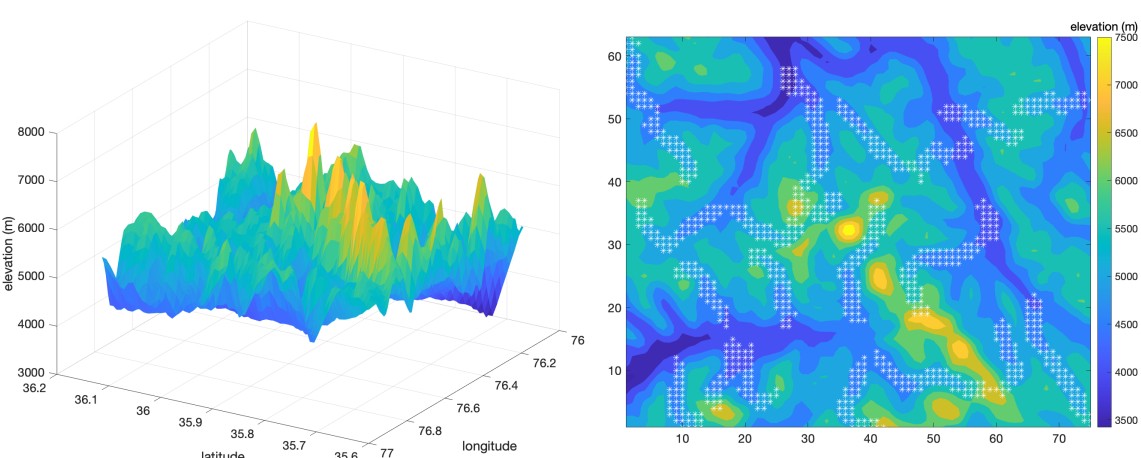

**Figure 2.** WRF innermost domain (D03) terrain elevation from GTOPO30 dataset, where the white shading indicates the additional glaciated regions derived using TopoToolbox (Schwanghart and Scherler, 2014).



**Table 1.** WRF model configuration.

| Model domains | |
| --- | --- |
| horizontal grid spacing | 8.5, 2.8, 0.9 km (D01-D03, as shown in Fig. 1a) |
| model top pressure | 10 hPa |
| vertical level | 50 |
| Model physics | |
| Microphysics | Thompson (Thompson et al., 2008) |
| Planetary boundary layer | MYNN2 (Nakanishi and Niino, 2006, 2009) |
| Surface layer scheme | Revised MM5 (Fairall et al., 2003; Jiménez et al., 2012) |
| Shortwave radiation | RRTM (Iacono et al., 2008) |
| Longwave radiation | RRTM (Iacono et al., 2008) |
| Land surface | NoahMP (Niu et al., 2011) |
| Lake physics | CLM 4.5 lake model (Gu et al., 2015) |
| Model forcing | |
| Run-time | February 1 - August 1, 2020 |
| Real-time data | 6-hourly NCEP GDAS/FNL 0.25 degree grid data |

## 2.1 Results

The modest glacier changes had an unexpectedly large-scale impact on regional summer precipitation and atmospheric circulation (July shown here for illustration). In addition to producing thermally-induced flow anomalies at the mountain-valley scale of the terrain (Fig. 3b) as we would expect from prior work (Vosper et al., 2018; Elvidge et al., 2017), adding missing glacier termini produced larger-scale, banded precipitation anomalies (Fig. 3a). This precipitation response averaged spatially to only a few millimeters over the innermost simulation domain (D03), but featured local anomalies exceeding 50 mm ($\approx 11\%$ change

relative to the July mean over D03), reflecting synoptic-scale shifts in the precipitation-generating flows that overwhelmed the effects of local topographically-induced circulations. The precipitation response moreover extended out beyond the inner domain, appearing as regional-scale banded anomalies exceeding 70 mm ($\approx 20\%$ change) (Fig. 3c) aligned parallel to the climatological southwesterly monsoonal moisture flux (arrows, Fig. 3d). Fig. 3b highlights the prominent imprint of mountain waves on the associated response in upper tropospheric moisture flux convergence. Notably, the robust synoptic-scale mois-

ture flow response exerted a substantial influence on the precipitation and overwhelmed the localized orographic disturbance, highlighting the significant impact of glacier ice on the monsoonal circulation and, hence, precipitation. These regional-scale, large-magnitude shifts in precipitation in turn would superimpose on snow distribution, glacier mass balance, and hydrology of the Karakoram range. For example, changing the distribution of snow affects both the mass accumulation (how much snow is added to the glacier system) and the mass ablation (melt) of a glacier. Indeed, summer snow distribution on glacier surfaces can

cause extremely large shifts in the surface energy balance through surface albedo feedbacks (Johnson and Rupper, 2020). Thus a small shift in snow distribution in the summer can have a large impact on both the glacier mass balance and the meltwater



production. It is therefore critical to quantify the influence of glaciers on atmospheric systems and the resulting impacts on glaciologic and hydrologic systems. Without this information, it is difficult to know when and where the glacier-atmosphere influence is significant, and how wrong our current estimations of mountain weather, climate, glaciers, and hydrology may be.

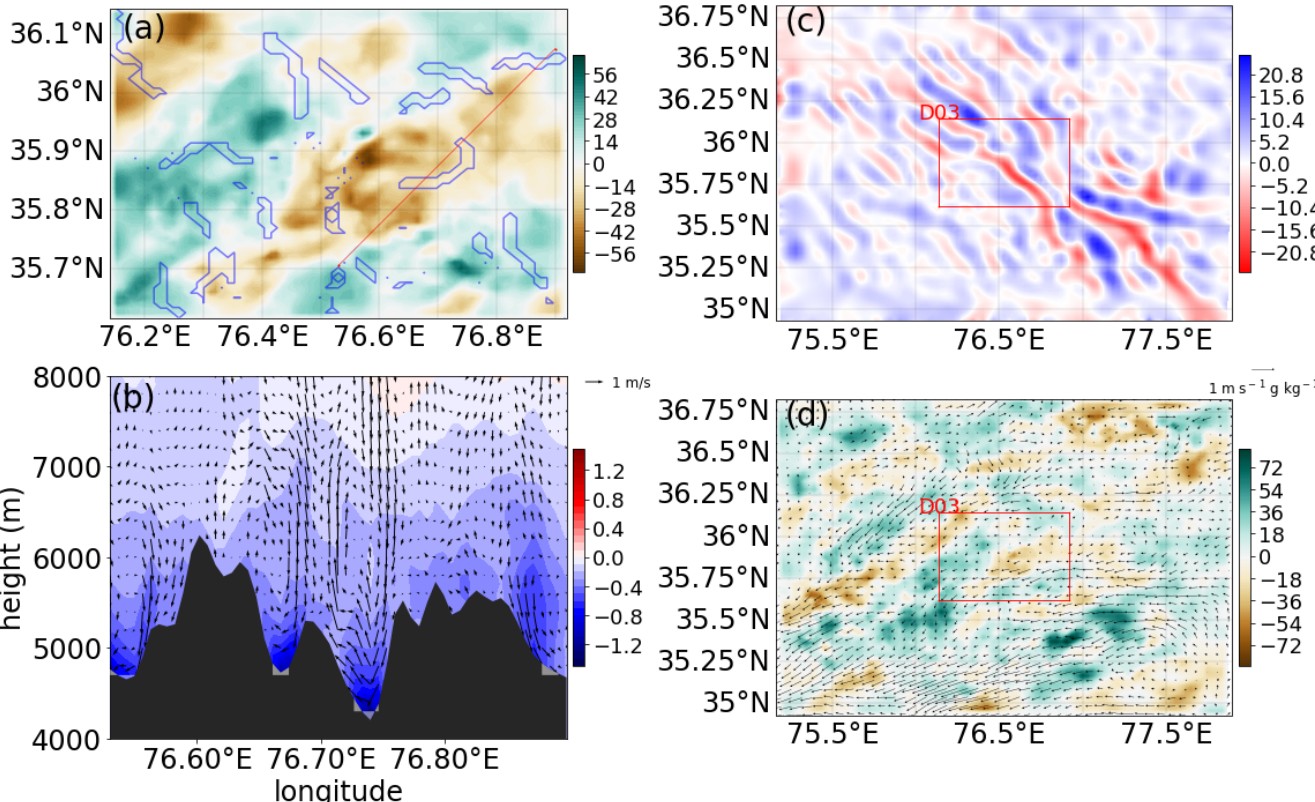

**Figure 3.** Simulated response of the atmosphere to the addition of glacier extent indicated by blue shading in Fig. 1c during July 2020. (a) Response of precipitation (mm/month) in the innermost domain (D03). (b) Response of wind velocity (vectors, m/s) and potential temperature (shading; K) along the red transect in (a), with terrain shaded black and added glaciers shaded gray (in valley bottoms). (c) Response of 350-hPa moisture flux convergence (shading; $g \cdot kg^{-1} \cdot s^{-1}$) on D02, with red box indicating D03. (d) Response of 350-hPa moisture flux (arrows; $m \cdot s^{-1} \cdot g \cdot kg^{-1}$) and precipitation (mm) on D02 with red box indicating D03.





## 3 Conclusions

Impacts of mountain glaciers on atmospheic circulation and precipitation are potentially significant, but glaciated regions are often incorrectly specified in default land surface fields or are lost as subgridscale features in coarser models. The state-of-art, high resolution numerical weather prediction model, Weather Research & Forecasting (WRF) Version 4.5, was used here to investigate the atmospheric circulation response to alpine valley glaciers. The missing valley glaciers and termini were filled in with an idealized form derived from the digital elevation model provided in WRF, and the lower-elevated valley regions are defined using TopoToolbox, denoted as the glacier-adjusted area. In this region, we modified the landuse category to classify it as glaciers (snow-ice) to effectively force the model internal land surface to represent glaciated terrain. However, despite the improvements in land surface representation, WRF's treatment of glaciers remains simplified, as it does not explicitly simulate glacier mass balance, ice dynamics, or detailed subgrid-scale processes, potentially leading to biases in land-atmosphere coupling.

Numerical model simulations highlighted that the presence of valley glaciers plays a substantial role in influencing the near-surface wind pattern. Particularly, extension of glaciers to lower elevations has a significant weakening effect on the up-valley flows, leading to a pronounced strengthening of the descending motion within the valley. This indicates the enhancement of the katabatic winds, often referred to as glacier winds, due to the presence of valley glaciers. Additionally, the precipitation response pattern is dominated by synoptic-scale features that overwhelm the effects of the valley-scale anomalous circulations.

The glaciated area added in the experiment impacted the atmosphere through localized changes in albedo and surface turbulent heat fluxes. However, these changes aggregated regionally to induce a translation of the predominant monsoonal precipitation features along a northwest-southeast axis, i.e., perpendicular to the monsoonal flow. Simulations of hydroclimate that misrepresent retrospective or future glacier extent may thus have unknown, large-scale banded precipitation errors of sufficient magnitude to impact glacier mass balance, precipitation-related hazards, ecosystem dynamics, and hydroclimate more broadly. As an additional implication, the multi-scale response of the atmosphere to glaciated area indicates an important feedback process that will influence how hydroclimate evolves in response to projected glacier retreat. Thus, the results from the Karakoram region highlight the need to assess further how the presence of glaciers impacts regional-scale weather and climate in differing topographic and climatic settings. Neglecting to include glaciers may result in large errors in hydroclimate at scales relevant to estimating downstream water resources, probability distribution of hazards, hydropower potential, and myriad other processes of societal concern.



*Data availability.* Meteorological boundary forcing data for WRF configuration is available at https://rda.ucar.edu/datasets/ds083.3/ (National Centers for Environmental Prediction, National Weather Service, NOAA, U.S. Department of Commerce, 2015). Model output data is archived in Utah Center for High Performance Computing (CHPC) storage space and available upon request.

*Author contributions.* HB, SR, and CS conceived the project idea; HB, SR, and CS designed the research; HB and CS performed the research and analyzed data; and all authors contributed to writing the paper.

*Competing interests.* Authors declare that they have no competing interests.

*Acknowledgements.* This project was supported by National Science Foundation award FRES 2218664 and NASA 19-HMA19-0029. We thank the University of Utah Center for High Performance Computing (CHPC) for computational resources and University of Utah IT for
computer-support services.



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
