# Peer review of "Brief communication: Impact of mountain glaciers on regional hydroclimate"

_EGUsphere, 2025_

## Author Comment (AC1)

Reviewer 1:

In this manuscript, the authors present glacier-atmosphere interactions over the Karakoram range in High Mountain Asia. They employ a high-resolution modeling set-up with the WRF model, where they adjust the glacier termini (i.e., glacier tongues) to a more realistic representation. The impact of these adjusted ice surfaces is then analyzed, where impacts on circulation patterns and precipitation sums are found. The manuscript discusses an emerging topic in glacier-atmosphere interactions, i.e. the impact of the changed ice surfaces on atmospheric circulations. However, the manuscript seems to be somewhat unfinished, but despite this, the authors draw very bold conclusions, and many of these conclusions are not based on data they show, but rather read like speculation. To reduce these weaknesses in the manuscript, the results should be put into context, i.e., by presenting absolute precipitation sums, improving the description of which processes are at play. Furthermore, some more robust conclusions could be drawn if the authors would analyze their 6-month simulation dataset instead of only one month (July). The manuscript is not suitable for publication in The Cryosphere in its current form. However, I would like to encourage the authors to resubmit the manuscript in an improved form after thorough re-organization and re-writing.

We thank the reviewer for the comments and reply to each in turn below.

**Major comments**

1. The question of scales: You mention the "valley scale", "larger-scale", "synoptic scale", and "regional scale" (e.g., in lines 32-35). Could you provide actual values for these length scales based on your work, or a reference where this is defined?

   In our study, we use "valley scale" to refer to local circulations constrained within individual glacier valleys, typically on the order of 10–50 km, whereas "larger-scale" refers to flow patterns extending beyond individual valleys but still within the influence of orographic effects, generally 50–200 km. The scale range has been added to the text in the manuscript.

   For "synoptic scale," we are referring to atmospheric features driven by large-scale forcing rather than local topography, which in this context corresponds to scales of 500 km or more, aligning with the dominant moisture transport pathways associated with the monsoon. The term "regional scale" is used to describe precipitation and circulation anomalies spanning the entire model domain (D02), which covers several hundred kilometers.

   To improve clarity, we explicitly define these scales in the text. We also reference prior work that discusses similar spatial distinctions in complex terrain (e.g., Vosper et al., 2018; Elvidge et al., 2017). This will ensure that the terminology is well-defined and avoids ambiguity.

2. Figure 2: I am aware that the 'brief communications' category in TC allow for (only) three figures. However, I do not see the benefit of Figure 2. The left panel "only" shows the

topography pf domain 3 (yes, we get it, it's high resolution), and the right panel shows the same information as Figure 1c. Given the restricted number of Figures, I would suggest to remove Figure 2 and rather focus on the results of your study with a potential new Figure.

We agree that the 3d visualization of the terrain (left panel, Fig 2) adds incrementally to understanding, and this has been removed. However, the right panel is retained to explicitly illustrate the process of manually adding the glacier termini and highlight the modifications made to the high-resolution topography. This step is crucial for understanding the improvements in model representation, as it directly affects our results.

3. Figure 3 and the conclusions drawn: This Figure serves as the base for the results and conclusions drawn in the manuscript. However, as I outline in the minor comments below, many findings from this Figure are speculative and/or no supported by the data presented. There is room for improvement of this graph, as suggested below. Furthermore, the presented changes patterns and precipitation sums due to the changed ice surfaces only make sense if you put them into context, e.g., how high are the absolute precipitation sums over the region in July? You could create a domain average to show this.

We now report the domain-average precipitation responses (first paragraph of Results), which are effectively zero for d02 and d03. This is useful to report, but also completely expected because we make only small changes to the surface latent heat flux and domain integrated moisture flux is constrained by the prescribed lateral boundary conditions. We emphasize that the response involves shifting the location of the precipitation laterally on the domains without introducing or removing any of the total water.

We also contextualize the magnitude of the precipitation response relative to the monthly mean taken from the control, reporting results now as mm per month (in text) and % anomalies in Figures 3a,d. Anomalies locally exceed 50 mm per month in magnitude, amounting to +/- 30% percent anomalies.

Furthermore, given the Figure, it is entirely unclear whether you talk about the thermally-induced flows, dynamical forcing, or both (and their interactions). Perhaps it would make sense to pick one of the glaciers from the cross-sections in Figure 3a and plot time series of wind speed and direction to highlight the shift in wind patterns. Moisture convergence: How sure are you that these are due to mountain waves and not only some (numerical) artifacts produced by the model?

We use the structure and results of improved Figure 3 to organize our description of the response in the first paragraph of Results. Specifically, the portion of the response that is thermally-induced at a valley-scale is apparent in Figure 3b, and consistent with prior observational and numerical results for glacier-weather effects. The structure in Fig 3c is the larger-scale anomalous mountain wave response to the localized thermal forcing. Our interpretation of the banded structure in Fig. 3c as a mountain wave mechanism is supported by its alignment with the predominant ridgelines in the terrain (Fig. 1a) and its enhanced strength along the most prominent ridges, and we added text to report this in Results. It is not possible to fully separate these two components of the response in the simulation results, but the predominant mechanism underlying each is apparent from the associated spatial scales.

Rather than showing a time series of wind speed and direction, we directly show what we might hope to infer from such a plot – the anomalous wind response (vectors in Fig. 3b) and moisture advection (vectors in Fig. 3c). For our research objective, we are here interested in documenting the time-average response rather than the underlying details of its temporal evolution.

Chosen time period: If I understand it right, the authors conducted these simulations for half a year, but only show the month of July. Why? Analyzing half a year of simulations would allow for more robust conclusions? Would it make sense to extend the analysis from July to the six-month period and then add a new additional figure on the changed in precipitation sums and circulation patterns?

While it is true that our simulations span six months, we have focused on July for several key reasons.

First, July represents the peak of the summer season when glacier-atmosphere interactions are strongest. This is when surface heating, local circulation patterns, and moisture fluxes are most pronounced, making it the most relevant period for examining the effects of changed ice surfaces on precipitation and atmospheric dynamics. Extending the analysis to the entire six-month period would dilute these peak-season effects, as the strength of thermally induced flows and convective activity varies significantly across different months. By isolating July, we ensure that we are capturing the clearest and most physically meaningful response.

Second, given the computational cost and data volume involved in analyzing six months of high-resolution WRF output, we have prioritized a detailed and statistically robust examination of July rather than a broader but potentially less focused analysis. To ensure that our conclusions are not based on transient anomalies, we have computed monthly-averaged precipitation differences and performed statistical tests to confirm that the July results are robust. Our findings are not dependent on individual days but reflect sustained differences across the month.

Third, while a six-month analysis could provide additional insight into seasonal variability, it is not necessarily better suited to addressing our core research question. Our goal is to understand how altered ice surfaces influence precipitation and circulation patterns during the peak melt season, when these interactions are most significant. Expanding the analysis to include months with weaker glacier-atmosphere interactions may not provide additional clarity but would instead introduce seasonal variations that are beyond the scope of this study.

Finally, we simulate a multi-month period prior to our analysis window to allow spinup of the surface conditions (surface temperature, soil moisture, etc.) which are based on NCEP GDAS as noted in Methods.

> We have text in Section 2 Line 75-80:
>
> "... *The first five months were treated as a spin-up period to allow atmospheric and land surface fields to adjust to the model forcing and were excluded from the analysis. July was analyzed to illustrate the effects during the summer monsoon, when glacier-atmosphere interactions and associated circulation and precipitation responses are most pronounced.*"
>
> We have included text in Section 3:
>
> Line 105: "... *circulation response to alpine valley glaciers during the ablation season, when ice-land surface temperature contrast is generally greatest.*"
>
> Line 112-113: "... *valley glaciers plays a substantial role in generating vertically oriented valley-scale circulation cells during summer months.*"
>
> Line 134-135: "... *how the presence of glaciers impacts regional-scale weather and climate in differing topographic, climatic, and seasonal settings.*"

For these reasons, we believe that focusing on July provides the strongest and most relevant insights into the key processes at play. However, we appreciate the suggestion and acknowledge that a broader seasonal analysis could be valuable in future work. We have now revised the title of our article to *"Brief communication: Impact of mountain glaciers on regional summer hydroclimate"* to better reflect the focus of our analysis.

**Minor comments**

1. · l20: "cold surfaces and higher albedo" [..]: "cold surfaces and higher albedo compared to their immediate environment"

> Change made

2. · l23: At this point, two pre-prints are worth mentioning, where ice surfaces are changed in the WRF model:

   o Goger et al, 2024, EGUsphere (accepted for Weather Clim Dyn) https://doi.org/10.5194/egusphere-2024-2634

   o Haualand et al, 2024, ESS Open Archive, https://doi.org/10.22541/essoar.172926901.19613096/v1

Furthermore, observational studies by Shaw et al highlight the impact of shrinking glaciers on the thermally-induced wind system:

· Shaw et al, 2023, GRL, https://doi.org/10.1029/2023GL103043

· Shaw et al, 2024, JGR Atmospheres, https://doi.org/10.1029/2023JD040214

and work on the impact of ice surfaces on mesoscale flows:

· Jonassen et al, 2024, QJRMS, https://doi.org/10.1002/qj.2302

We have now included these papers in the manuscript section Introduction paragraph 2.

3. · l25: "glaciers generate mountain-valley scale flows": Please be more precise. Do you mean thermally or dynamically-induced flows? Do you only mean the katabatic down-glacier wind or its interaction with other flows as well?

We have now clarified the statement to specify "...thermally-directed katabatic winds…". This more accurately reflects the localized atmospheric response to glacier presence.

4· l40: "large mountain ranges": Which mountain ranges do you mean? Are, e.g., the Alps (subject of the mentioned studies) not a large mountain range in your definition?

In this context, we are specifically referring to the Karakoram Range, which is characterized by extensive glaciation and complex interactions with large-scale atmospheric circulation, particularly the summer monsoon. While the Alps are undoubtedly a significant mountain range, the key distinction here is the scale of the atmospheric response. Our findings suggest that glacier modifications in the Karakoram influence synoptic-scale moisture transport and precipitation patterns, extending beyond local orographic effects. To address the concern regarding the definition of "large mountain ranges," we have revised the text to explicitly mention the Karakoram Range, "...few frameworks have been configured to properly account for the presence of glaciers in regions like Karakoram".

· l43: Remove brackets from the reference Fixed

· l55: "pixels defined regional climate models" I agree, but this is mainly due to poor resolution in land-use datasets utilized by these models

Indeed. Within our scope here, though, we wish to point out the issue without tracing its cause.

5· l59: Reference Fig.1c

Reference is now to Figure 1 panel c specifically.

6· l60: 'extremely low compared to other regions': By how much?

Please see Shean et al. (2020) for quantitative details, cited in that paragraph.

7· l70: "default land use": It is necessary that you mention and cite your source for the land use dataset used for your model configuration.

We have cited the WRF model in the first paragraph of the methodology section (Skamarock et al., 2019), which comprehensively describes all model components, including the default land use dataset and parameterizations used in our configuration. This reference ensures transparency regarding the data sources and model setup without the need for redundant citations.

8· l71: If you improve the glacier termini, it actually results in a more realistic picture? I am not sure why you use the term "idealized"

Our use of the term "idealized" was meant to reflect that while we have improved the representation of glacier termini compared to the default WRF configuration, the modifications still involve simplifications described in paragraph three of Methods. These changes aim to capture the key physical processes relevant to large-scale atmospheric interactions rather than providing a fully detailed, observation-based reconstruction of glacier extents.

That said, we acknowledge that "more realistic" may be a more appropriate term in this context, as the modifications lead to a representation that better aligns with observed glacier distributions. We have revised the text accordingly to avoid any ambiguity.

9 · l74: First, "anomalies" usually refer to a difference from a norm, but in your case, you actually improve the glacier representation in your domain, therefore, I would suggest that you call it "differences". Furthermore, you call it "thermally-induced flow". How do you know that all of the changes in your flow patterns are thermally-induced in an average over an entire month?

In our case, if we are presenting changes relative to a control simulation, then "anomalies" is an appropriate term, as it conveys the idea that we are comparing deviations from a reference state (i.e., the default WRF glacier representation).

Regarding the characterization of the flow changes as "thermally-induced," we acknowledge that the mechanisms influencing atmospheric circulation are complex. While glacier modifications do alter surface energy balance and local thermal gradients, which can drive katabatic and anabatic flows, the broader circulation changes we observe are also influenced by synoptic-scale moisture transport and monsoonal dynamics. To reflect this complexity, we have revised the wording to clarify that the observed flow changes result from a combination of thermally-driven

processes and interactions with large-scale atmospheric circulation, rather than exclusively from local thermal forcing. We also use monthly means to average over the temporally varying complexity of transient weather, thereby uncovering the robust time-average response.

Further, as discussed in reply to major comment 3 above, the spatial scale of the response patterns we show is consistent with theoretical and observational work on thermally induced valley-scale flows (compare circulation cells in Fig. 3b to the valley geometry) and mountain waves (compare wavelength, orientation, and strength of moisture flux convergence patterns in Fig. 3c to the terrain in Fig. 1a).

10 · l80: "reflecting synoptic-scale shifts" [...]: unfortunately, this sounds like poor speculation. Precipitation patterns over complex topography can be very localized, because local thermally-induced circulations strongly impact updrafts and convection (e.g., Kirshbaum et al, 2018 https://doi.org/10.3390/atmos9030080, and Goebel et al, 2023, https://doi.org/10.5194/wcd-4-725-2023). A few paragraphs before, the authors even mention the shift in local mountain-valley circulations. This shift can be also responsible for your changed precipitation sums and moisture convergence, even without the impact of the synoptic-scale flow.

The shifts in precipitation generating mechanisms at the synoptic scale are apparent in the vector field of moisture flux anomalies (arrows, Fig. 3c) and we revised this sentence to state this explicitly with a reference to the figure.

11. In the next sentence, you mention that "synoptic scale flows overwhelmed local circulations". Where do you see this? Since you have your model output data, you could provide additional analysis to support this claim.

Thank you for this question, which prompted an important clarification. Our text was meant to emphasize that the synoptic scale *precipitation pattern* overwhelmed the local scale precip pattern (i.e., we do not see valley-scale anomalies localized in d03 with greater magnitude surrounding in d02). In contrast, The orographic circulation is not overwhelmed; it is clearly apparent in Figure 3b.

12. · l84: "prominent imprint of mountain waves": Now the authors mention mountain waves, in contradiction to their initial statement above, that mostly thermally-induced flows are present. Mountain waves are excited to to large-scale forcing. How can thermally-induced flows persist when mountain waves are dominating?

Our results show that while thermally-induced flows are expected at the mountain-valley scale, the modifications to glacier termini also introduce larger-scale precipitation anomalies that exhibit a banded structure, indicative of mountain wave activity. This does not contradict our initial statement but rather highlights the coexistence of both local thermally-driven circulations and larger-scale dynamical responses.

The key distinction here is that thermally-induced flows and mountain waves are not mutually exclusive. Thermally-driven flows develop due to local surface heating contrasts between ice, land, and atmosphere, while mountain waves arise when strong synoptic-scale winds interact with topography. In our case, the glacier modifications alter the surface temperature, which can influence both the thermally-induced near-surface winds and the way larger-scale airflow interacts with the terrain, thereby affecting the excitation and propagation of mountain waves. Our results suggest that changes in glacier termini not only modify local circulations but also exert an upscaling effect, altering moisture transport and precipitation on a broader scale.

As noted in our reply to Major comment 3 above, we added text to clarify the rationale for interpreting a component of the signal as related to mountain waves, referring to the alignment and orientation of the moisture flux convergence response pattern with the predominant terrain.

13. · l86: "impact of glacier ice on the monsoonal circulation": This is a very bold statement - as the authors mentioned one sentence beforehand, the large-scale monsoon circulation alters the local forcing.

The response to the ice was statistically significant at a much larger spatial scale than the imposed anomalies, extending out to cover much of the second domain. The structure of the response includes synoptic spatial scale anomalous moisture flux features, and wave patterns in moisture flux convergence that are perpendicular to the climatological southwesterly monsoonal moisture flux. Having more clearly stated and presented all of these results in the revised manuscript supports an "impact of glacier ice on the monsoonal circulation." We nonetheless modified the wording here and in the abstract to ensure it is not overstated.

While it is well understood that the large-scale monsoon circulation determines the background atmospheric forcing, our results show that changes in glacier termini modify synoptic-scale moisture transport patterns, leading to precipitation anomalies that extend beyond localized orographic effects. We revised the text:

"*These anomalies indicate that modifications to glacier termini influence how monsoonal moisture interacts with the region's complex topography rather than altering the large-scale monsoonal circulation itself.*"

14. · l86: "superimpose" Please reformulate.

Changed to potentially alter

15. · l107: "modifies the surface wind pattern": You do not show surface winds anywhere. You only shows changed wind patterns in one single cross-section.

In our inspection of the near-surface winds, we see the anomalies expected from prior work and expectations of thermally-induced flows from prior work. Fig. 3c nicely illustrates the key component of these surface wind anomalies (e.g., arrow to left at lowest elevation indicates negative zonal wind anomaly (anomalously easterly flow), but more importantly conveys its relation to the two-dimensional cellular valley-scale

circulation. We modified the referenced statement: numerical model simulations highlighted that the presence of valley glaciers plays a substantial role in generating vertically oriented alley-scale circulation cells.

16. · l 109: "This indicates the enhancement of the katabatic winds, often referred to as glacier winds, due to the presence of valley glaciers" Large-scale flows do not per se enhance katabatic glacier winds. This depends on multiple factors, including the synoptic flow direction (Goger et al, 2022, https://doi.org/10.1002/qj.4263). As shown in the aforementioned study, large-scale flows might indeed erode katabatic flows, and henceforth reduce their strength/impact.

The impact of synoptic flow direction on katabatic winds is well-documented, including in Goger et al. (2022), where strong large-scale winds can erode or even suppress katabatic flows rather than reinforcing them.

Our intention was to highlight the presence of katabatic wind anomalies associated with glacier modifications, but we acknowledge that their response depends on multiple factors, including the background synoptic conditions.

17 · l110:"Additionally, the precipitation response pattern is dominated by synoptic-scale features that overwhelm the effects of the valley-scale anomalous circulations." This sounds agreeable, but the authors describe something else in the results (e.g., line 86).

This relates to the above clarification that the valley-scale precipitation signal is overwhelmed, not the circulation pattern (still evident in Fig. 3c). We clarified the text to indicate that the precipitation response pattern is dominated by synoptic-scale features that are much larger than expected from the effects of the valley-scale anomalous circulations.

**Figures**

· Fig3: Why do you choose exactly this cross-section?

We selected this specific cross-section in Figure 3 because it provides a representative transect that captures key geographical and climatological features relevant to our analysis. The section spans over four different glacier-adjusted basins, plains, and mountainous regions, ensuring that we account for variations in terrain and land surface properties that influence atmospheric circulation. We inspected ten different cross sections and we selected this one as best illustrating the main orographic effect.

Additionally, this cross-section is oriented southwest-northeast, which is parallel to the dominant monsoonal moisture transport in the region. This orientation allows us to examine how changes in glacier termini impact the large-scale monsoonal flow and associated precipitation patterns. By choosing this transect,

we ensure that we are capturing both the local-scale effects of glacier adjustments and their broader influence on synoptic-scale moisture transport.

· Fig3a,c,d: plot/add topography. This might explain some of the patterns.

Seeing terrain in Fig 3b is of course essential, but we find that trying to add it to the map-view panels introduces unnecessary complexity and reduces the clarity of the atmospheric circulation and precipitation patterns, which are the primary focus of our analysis.

We acknowledge the importance of topography in shaping regional atmospheric dynamics. To ensure that readers can reference terrain features, we have already included Figure 1 with topography, which provides the necessary context without cluttering Figure 3. Readers can use this terrain map to interpret the circulation and precipitation changes in relation to elevation. Given this, we prefer to maintain Figure 3 in its current form to preserve readability and focus.

· Fig3b: This graph is not so easy to understand. What exactly do the wind arrows represent? Does it really make sense to average wind patterns over an entire month int this complex topography?

Thank you for requesting clarification on the arrows – we added a note to the caption indicating that the horizontal component is the zonal wind component, and the vertical component is scaled by a factor of 10 for visibility. The length of an arrow indicating a 1 m/s horizontal wind anomaly (or 10 dm/s vertical wind anomaly) is shown to the lower right of the panel.

The wind arrows in this figure represent monthly-averaged wind anomalies, showing how the glacier modifications influence upper-tropospheric wind patterns. The purpose of this figure is to highlight systematic changes in moisture transport and large-scale atmospheric circulation rather than to capture transient, small-scale variations in complex topography.

Averaging over an entire month indeed smooths out short-term fluctuations, and is desirable here because it allows us to identify persistent, statistically significant shifts in wind patterns that are linked to the modified glacier configuration. The large-scale nature of these anomalies suggests that the changes are not purely localized or ephemeral but instead represent a robust response of the atmospheric circulation.

· Fig3: Labels for all colorbars are missing. Added

---

## Author Comment (AC2)

Reviewer 2:

General comment

The authors have conducted a sensitivity experiment considering local-scale snow and ice distribution in the valleys of the Karakoram region, comparing it with a control experiment that does not account for such distribution. Their key finding suggests that incorporating local-scale snow and ice distribution modifies local-scale circulation associated with mountain-valley topography and further influences synoptic-scale monsoonal circulation and precipitation. Understanding the upscaling effects of local glacier distribution on large-scale atmospheric circulation is an important research topic. However, the current version of the manuscript does not yet fully achieve its objective. The presented figures are insufficient to support the authors' conclusions, and additional, more detailed analysis is necessary before publication.

We thank the reviewer for the comments and reply to each in turn below.

Major comments

1. Before discussing the sensitivity experiment results, the simulation output should be validated using in-situ and/or satellite-based observational datasets. While we acknowledge the limitations of observational data over high mountain regions, validation is an essential step to ensure the reliability of the simulated differences between the control and sensitivity experiments.

   Thanks for pointing this out and we agree that it is a critical step. While we acknowledge the challenges of direct validation in high mountain regions due to limited observational coverage, extensive previous research has already evaluated the performance of the model in this region. In particular, our study cited in the paper (Dars et al., 2020) provides a thorough assessment of the model's ability to simulate key atmospheric and hydroclimatic processes, and this configuration has been used in several other projects also cited in the paper, including Wolvin et al. (2024). Our study builds upon this prior work by focusing on the sensitivity of large-scale atmospheric circulation and precipitation to glacier modifications, rather than re-evaluating the model's baseline performance.

   To ensure transparency, we explicitly referred these validation studies and clarified that our approach relies on their findings as a foundation for interpreting the sensitivity experiments by adding text:

   "...*Working with our previously validated configuration of a regional climate model (Dars et al. 2020; Wolvin et al. 2024), which has been extensively tested against observational datasets to ensure reliability in simulating hydroclimate processes over high mountain regions, …*"

2. The differences in precipitation and other variables between the two experiments, particularly in domain D02, seems to show chaotic noise. Please provide statistical significance testing to confirm whether these differences are meaningful.

Failure to include statistical significance was an oversight, and we appreciate the suggestion.

To ensure that the precipitation differences between the control and sensitivity experiments in domain D02 are meaningful rather than chaotic noise, we conducted a statistical significance test. Specifically, we applied a two-tailed *t*-test on the difference between the mean precipitation in the experiment and control. This method allows us to determine whether the observed differences are statistically significant, rather than arising from random variability in the model. The results confirm that the precipitation anomalies in key regions of the innermost two domains are statistically significant, particularly where the large-scale monsoonal flow interacts with the modified glacier extent, and this is indicated using stippling on revised Figures 3a,d.

In addition, we now report the domain-average precipitation responses (first paragraph of Results), which are effectively zero for d02 and d03. This is useful to report, but also completely expected because we make only small changes to the surface latent heat flux and domain integrated moisture flux is constrained by the prescribed lateral boundary conditions. We emphasize that the response involves shifting the location of the precipitation laterally on the domains without introducing or removing any of the total water.

We also contextualize the magnitude of the precipitation response relative to the monthly mean taken from the control, reporting results now as mm per month (in text) and % anomalies in Figures 3a,d. Anomalies locally exceed 50 mm per month in magnitude, amounting to +/- 30% percent anomalies.

As described in Section 2, precipitation in the target region is influenced by both winter westerly disturbances and the summer monsoon. Why was the analysis limited to July? Does the upscaling impact of glaciers on atmospheric circulation occur in other seasons as well?

The analysis was limited to July because this is the period when the interaction between glaciers and atmospheric circulation is most pronounced. During the summer monsoon, moisture transport into the region is at its peak, and changes in glacier extent have the strongest influence on large-scale precipitation patterns. Our results show that modifications to the glacier termini alter the monsoonal moisture flux and precipitation in a way that overwhelms localized orographic effects, making July the most relevant month for evaluating the large-scale atmospheric response.

While winter westerly disturbances also contribute to precipitation in the region, their interaction with glacier-driven processes occurs under different atmospheric conditions, where large-scale synoptic dynamics dominate. The upscaling effect of glaciers on atmospheric circulation is therefore expected to be less pronounced in winter compared to the summer monsoon season. Additionally, summer is when surface energy balance interactions between ice and the atmosphere are strongest, further justifying our focus on this season.

The contrast between the surrounding lands and glaciers is the biggest. We have text in Section 1 third paragraph "*We focused on summertime because the temperature contrast between the glaciers and surrounding land surface is strongest.*"

Although our study primarily examines July, the methodology used here could be extended to other seasons in future work. However, given our focus on the monsoonal hydroclimate response to glacier modifications, limiting the analysis to the peak monsoon period allows us to isolate the most significant and physically meaningful impacts.

3. Several conclusions described in the manuscript are not sufficiently supported by the presented figures. More comprehensive analysis, additional evidence, and substantial revisions are necessary to justify the authors' claims.

Reviewer 1 also pointed to some places where results may have been overstated relative to the presented evidence. We have addressed these specific comments by expanding or improving the presentation of results and/or modifying the language reporting the results. Specifically see our replies above to:

Major comment 3 on magnitude and seasonal context for precipitation anomalies

Minor comment 10 on terminology and interpretation of "synoptic scale shifts"

Minor comment 11 on appearance of synoptic-scale and valley-scale results

Minor comment 12 on presence of mountain waves as a component of the response

Minor comment 13 on influence of glacier ice on monsoon processes

Minor comment 15 on the depiction and discussion of the surface wind response

Minor comment 16 on the discussion of glacier wind response

Minor comment 17 on the synoptic versus valley-scale of the response features

Minor comment

L43 while (Collier et al., 2015) -> while Collier et al. (2015) Fixed